# Open6DOR: Benchmarking Open-instruction 6-DoF Object Rearrangement and A VLM-based Approach

Yufei Ding[1,2*]    Haoran Geng[1,2*]    Chaoyi Xu[2]

Xiaomeng Fang[3]    Jiazhao Zhang[1,3]    Songlin Wei[1,2]    Qiyu Dai[1]    Zhizheng Zhang[2]    He Wang[1,2,3†]

[1]*CFCS, Peking University*    [2]*Galbot*    [3]*Beijing Academy of Artificial Intelligence.*

https://pku-epic.github.io/Open6DOR

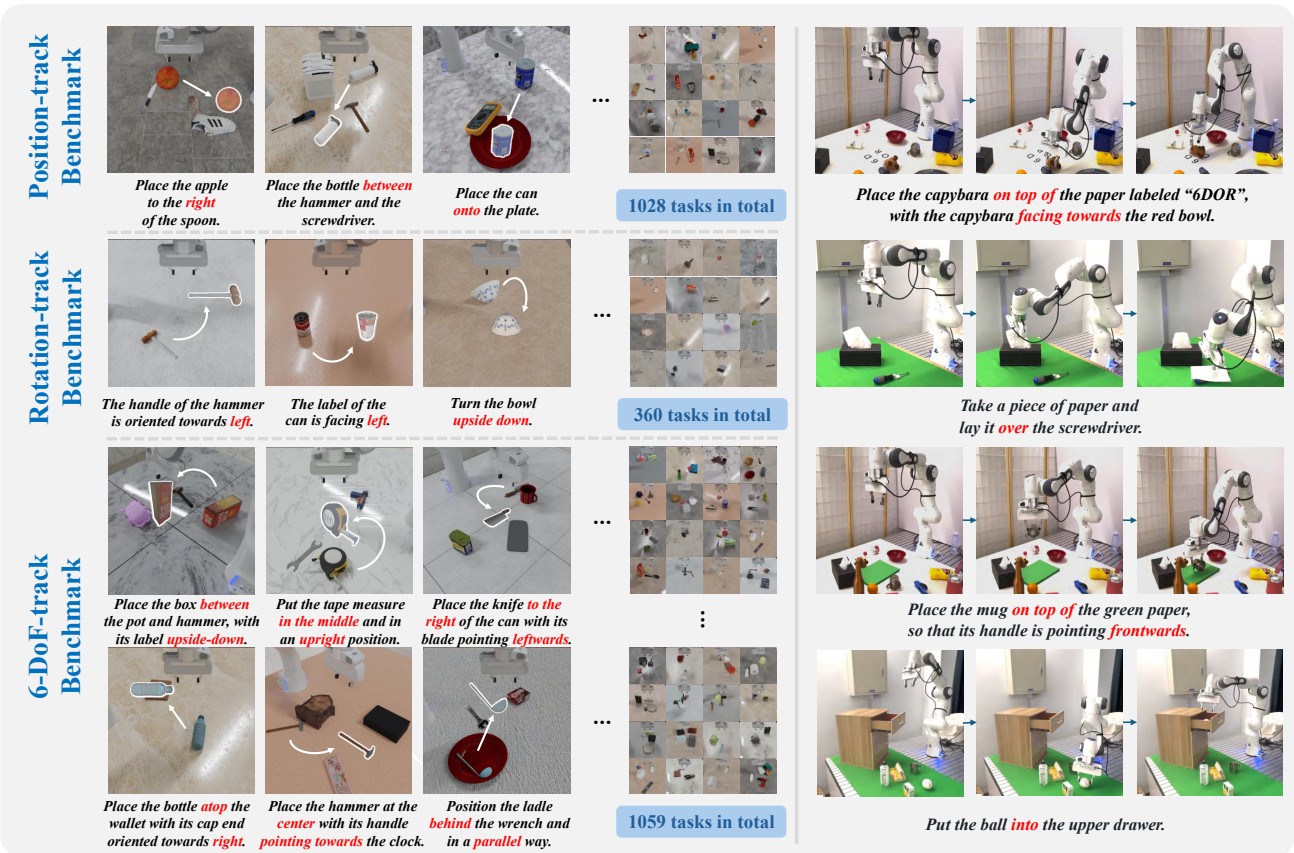

Figure 1. **Open6DOR Benchmark and Real-world Experiments.** We introduce a challenging and comprehensive benchmark for Open-instruction 6-DoF object rearrangement tasks, termed Open6DOR. Following this, we propose a zero-shot and robust method, Open6DOR-GPT, which proves effective in demanding simulation environments and real-world scenarios.

## Abstract

*The integration of large-scale Vision-Language Models (VLMs) with embodied AI can greatly enhance the generalizability and the capacity to follow open instructions for robots. However, existing studies on object rearrangement are not up to full consideration of the 6-DoF requirements, let alone establishing a comprehensive benchmark. In this paper, we propel the pioneer construction of the benchmark and approach for table-top Open-instruction 6-DoF Object Rearrangement (Open6DOR). Specifically, we collect a synthetic dataset of 200+ objects and care-*

*Equal contribution.

†Corresponding author: hewang@pku.edu.cn.

*fully design 2400+ Open6DOR tasks. These tasks are divided into the Position-track, Rotation-track, and 6-DoF-track for evaluating different embodied agents in predicting the positions and rotations of target objects. Besides, we also propose a VLM-based approach for Open6DOR, named Open6DOR-GPT, which empowers GPT-4V with 3D-awareness and simulation-assistance while exploiting its strengths in generalizability and instruction-following for this task. We compare the existing embodied agents with our Open6DOR-GPT on the proposed Open6DOR benchmark and find that Open6DOR-GPT achieves the state-of-the-art performance. We further show the impressive performance of Open6DOR-GPT in diverse real-world experiments. Our constructed benchmark will be released upon paper acceptance.*

## 1. Introduction

Large-scale multimodal models [9, 25] pre-trained on web-scale data have revolutionized numerous fields beyond what was previously imaginable, enabling open-vocabulary text understanding and 2D visual perception. The pursuit to bring general intelligence into the robotic realm and 3D physical world stands at an exciting yet nascent stage, calling for stronger capabilities in 3D-aware perception, robotic interaction and complex reasoning.

The advent of embodied large-scale models, exemplified by the RT series [2, 3, 7, 15] and VoxPoser [17], has demonstrated considerable progress in mobile or fixed-station pick-and-place operations. While these models are capable of rearranging the object positions following human instructions, they fall short of satisfying full 6-DoF object placement instructions that involve specified 3D rotations. This limitation renders them incompetent at many practical robotic applications, where both object position and orientation are essential. For instance, in our daily life we desire a water bottle to be placed upright, while on the shelves in retail stores, goods should face the same direction. Moreover, previous works[3, 15, 17] are often evaluated on their own robots in their own scenes with self-reported performance and nonstandard evaluation metrics. The absence of a standard evaluation protocol condone cherry-picking, obstruct comparative assessment, and thus, hinder the iterative enhancement of effective approaches.

In this paper, we target the task of Open-instruction 6-DOF Object Rearrangement, referred to as Open6DOR, which requires embodied agents to move the target objects according to open instructions that specify its 6-DoF pose. Open6DOR represents a fundamental skill for robotic manipulation tasks, presenting significant challenges in integrating instruction comprehension, 3D visual perception, and motion planning capabilities. Specifically, we promote the envelope of Open6DOR from two perspectives:

1) **Benchmark Construction:** We construct a standardized benchmark, namely Open6DOR Benchmark, which comprises 2447 tasks designed with 200 objects across diverse categories in simulation environments. For comprehensive evaluation, we divide the Open6DOR benchmark into the position track, rotation track, and 6-DoF track, each providing manually configured tasks along with comprehensive and quantitative 3D annotations. These tracks enable independent or combined translational, rotational, and overall performance assessments.

2) **VLM-based Approach:** We propose a VLM-based approach for Open6DOR tasks. Due to the challenges of Open6DOR analyzed before, all prior works, such as VoxPoser [17] and Dream2Real [20], fail to fulfill Open6DOR's 6-DoF requirements adequately. Most of them [17] determine the final positions of target objects while neglecting the rotation dimension. Among these efforts, Dream2Real[20] has the potential to consider position and rotation dimensions in a coupled way by utilizing a VLM to directly select the instruction-aligned result from all rendered images of the imagined rearranged scenes. This leads to almost intolerable time costs resulted from numerous renderings and VLM inferences, as well as unsatisfactory results due to the VLM's limited 3D perception, which renders it an incompetent critic. In contrast, we propose Open6DOR-GPT, which explicitly integrates 3D information from the initial scene to GPT-4V with equipped auxiliary modules and decomposes the translational and rotational determinations. In this way, we augment GPT-4V with 3D understanding capabilities and improve efficiency by reducing the determination space with decoupled modeling and simulation-assistance. Open6DOR-GPT achieves state-of-the-art performance in both benchmark evaluation and real-world experiments.

## 2. Related Work

### 2.1. Object Rearrangement Methods

Object rearrangement [1] requires an embodied agent to manipulate objects to the desired pose based on specific instructions. Early works [11, 12] address this challenge by using task-and-motion-planning (TAMP) which relies on pre-defined action primitives and close-to-perfect scene knowledge for trajectory sampling. TAMP methods are computationally inefficient and unscalable for complicated scenarios. To enhance generalizability and efficiency, recent research has shifted towards Learning-based approaches [10, 14, 24, 28, 33]. These methods are trained on simulators, predicting either high-level planning [10, 24, 28] or low-level actions [14, 33]. Despite exhibiting satisfactory performance in simulators, they suffer from a severe sim-to-real gap [24]. Recent advanced methods leverage the open-world understanding capabili-

ties of large language models (LLM) [6] or vision language models (VLM) [21, 26] for real-world deployment. A part of these approaches [16, 20] construct carefully designed prompts to describe the environments, and query off-the-shelf LLMs or VLMs for placement guidance before execution. Other methods [31] train large language models from self-collected data and directly output low-level actions. Departing from existing works that predominantly focus on location, our method also emphasizes rotation, leading to a 6-DoF rearrangement method.

## 2.2. Object Rearrangement Benchmarks

Benchmarking object rearrangement is extremely challenging and requires extensive annotation of ground truth placement. Existing object pick-and-place benchmarks [10, 32, 34] leverage pre-built environments such as Repilca-CAD [27] or hand-crafted scenes, and high-quality object reconstructions [4]. Besides, these benchmarks contain annotations about the initial position and target position of each object, which are leveraged for evaluating correct placement. Aside from the simulator benchmarks, there are real-world benchmarks [23] which directly evaluate baselines in real-world robots. However, even though these benchmarks can directly assess real-world performance, they fail to accurately replicate the testing environments for all baselines and are not permanently available due to hardware limitations. Different from existing works, we propose a 6-DoF object rearrangement benchmark that comprehensively evaluates 6-DoF placement, providing both position and rotation annotations.

## 2.3. Vision-Language Models for Open-instruction 6-DoF Tasks

Large models trained on internet-scale data have demonstrated great potential in high-level planning[13, 17? ]. The recent advent of VLMs further bridges the gap between visual perception and textual interpretation, empowering embodied agents with semantic understanding of scenes and instructions[19] to perform 6-DoF tasks. Some of the prior works[17] leverage VLMs to compose 3D value maps, planning robot trajectories that comply with the given instruction; while other methods such as Dream2Real[20] employ VLMs as evaluators, generating goal states in the form of images for VLM to assess. However, both approaches fail to consider complex tasks that strictly specify the rotation of an object. Moreover, Dream2Real suffers from excessive time-consumption and VLM's inaccurate judgment. In contrast, our method addresses the rotation and position aspect of the 6-DoF problem in a decoupled way, enhancing VLM's decision-making capabilities while expediting the inference process.

## 3. Open6DOR Benchmark

### 3.1. Task Formulation and Benchmark Overview

**Open6DOR task formulation.** Open-instruction object rearrangement refers to the process wherein an embodied agent repositions objects within a scene from an initial state, following specific instructions. In particular, a 6-DoF object rearrangement task focuses on repositioning objects in a 6-DoF space, including both orientational and translational movement. For a long-horizon rearrangement problem, we decompose the process into several independent pick-and-place tasks, during which objects are repositioned one at a time. We define each of these tasks as an Open6DOR task, in which a single target object is moved from its initial pose to a goal pose based on an open-vocabulary instruction. The input comprises a single-view RGB-D image of the initial scene captured by a camera fixed on the robotic arm, denoted as $I_{rgbd}$, along with an arbitrary task instruction $\tilde{I}$, which describes the desired goal pose of an object in the scene. Based on these, the model is required to output the quantitative goal position $P_{\text{goal}}$ and goal rotation $R_{\text{goal}}$ of the target object.

**Open6DOR benchmark.** The Open6DOR Benchmark is specifically designed for table-top Open6DOR tasks within a simulation environment. Our dataset encompasses 200+ high-quality objects, forming diverse scenes and totaling 2400+ diverse tasks, with statistics shown in Tab. 1. All tasks are carefully configured and accompanied by detailed annotations. To ensure comprehensive evaluation, we provide three specialized tracks of benchmark: the Rotation-track Benchmark $\mathcal{B}_r$, the Position-track benchmark $\mathcal{B}_p$, and the 6-DoF-track Benchmark $\mathcal{B}_{6DOR}$. $\mathcal{B}_r$ encompasses tasks achievable through a singular rotational movement at a fixed point—for example, "place the cup upside down". $\mathcal{B}_p$ concentrates on tasks requiring the repositioning of an object, like "put the cup between A and B", without specific regard to the object's orientation. Meanwhile, $\mathcal{B}_{6DOR}$ integrates both rotation and position requirements, involving tasks such as "place the mug in front of A with its handle pointing towards the left". Constructing the benchmark was a challenging and laborious task. The process involved four stages: a) data collection and processing, b) instruction design, c) task formulation, and d) pose annotation . It took the team over a month to complete the preliminary version, and we anticipate further investment to expand and refine the Open6DOR Benchmark.

### 3.2. Position-track Benchmark

**Data composition and annotation.** The Position-track benchmark includes 1028 tasks, each set in a table-top scene that contains several objects. Our synthetic object dataset $\mathcal{O}_s$ comprises 200+ items, covering a range of 70+ distinct categories. Originally derived from YCB[5] and Objaverse-

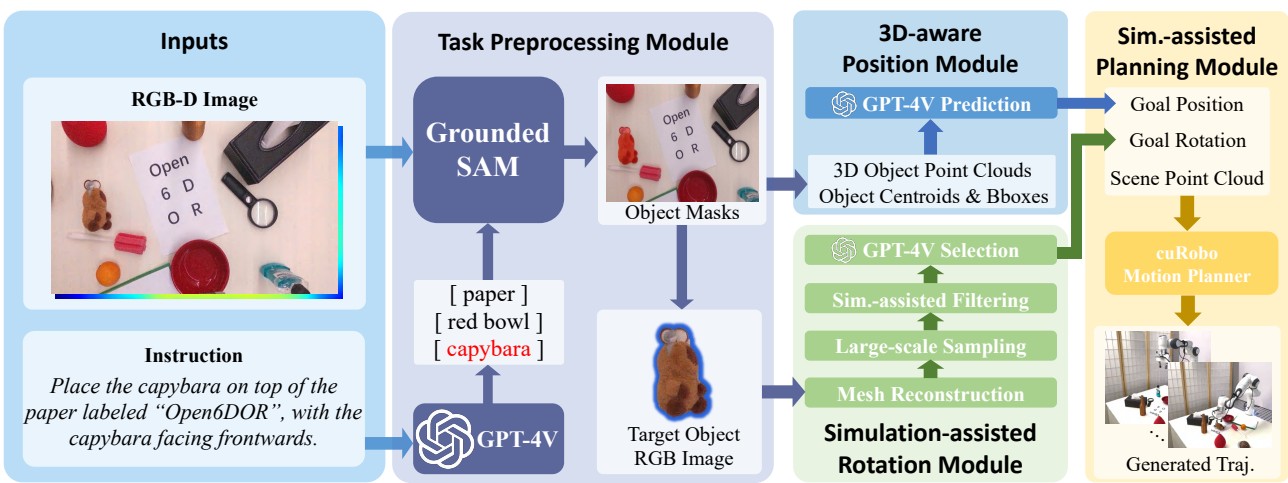

Figure 2. **Method Overview.** Open6DOR-GPT takes the RGB-D image and instruction as input and outputs the corresponding robot motion trajectory. Firstly, the preprocessing module extracts the object names and masks. Then, two modules simultaneously predict the position and rotation of the target object in a decoupled way. Finally, the planning module generates a trajectory for execution.

| Track | Position-track | | | | | | | | Rotation-track | | | 6DoF-track |
|---|---|---|---|---|---|---|---|---|---|---|---|---|
| Level | Level 0 | | | | | Level 1 | | Level 2 | Level 0 | Level 1 | Level 2 | - |
| Task Catog. | Left | Right | Top | Behind | Front | Between | Center | Customized | Geometric | Directional | Semantic | - |
| Task Stat. | 152 | 185 | 177 | 143 | 66 | 149 | 156 | 10 | 147 | 106 | 107 | 1059 |
| Benchmark Stat. | 1028 | | | | | | | | 360 | | | 1059 |

Table 1. **Statistics of Open6DOR Benchmark.** The entire benchmark comprises three independent tracks, each featuring diverse tasks with careful annotations. The tasks are divided into different levels based on instruction categories, with statistics demonstrated above.

XL[8], the objects are carefully filtered so as to ensure our selections are physically intact and semantically reasonable to be placed on a table. We normalize the scale of all the objects and use a uniform format of mesh representation. The objects are then classified into different categories for the convenience of future analysis.

For scene configuration, we randomly select 2-6 objects from $\mathcal{O}_s$ and position them on the table with random initial poses. We then generate an RGB-D image for each of the scenes and filter out low-quality ones (e.g. scenes that include unreasonably placed or heavily occluded objects), resulting in a single-view RGB-D image dataset $\mathcal{V}_p$. For position instructions $\mathcal{I}_p$, we design three levels that evaluate the understanding of: basic directions (Level 0) such as *Left, Right, Top, Behind, Front*; object relations (Level 1) such as *Between, Center*; and customized commands (Level 2) like *Put A into B*. The task instructions adhere to a uniform format, such as 'place A in front of B', where A and B are subsequently specified based on the context of individual scenes. Additionally, we annotate a position range for each task according to the given instruction.

**Evaluation metrics.** We assess the predicted goal postion

$P_{\text{goal}}$ according to the annotated position range. A position that falls into that range is considered as correct, otherwise wrong. For instance, in the *Left* task category, we verify whether the predicted position is to the left of the reference object (indicated by a smaller y-axis coordinate).

### 3.3. Rotation-track Benchmark

**Data composition and annotation.** The Rotation-track Benchmark consists of 360 diverse tasks, each set in a scene containing a single object. We use the same object dataset $\mathcal{O}_s$ as the Position-track Benchmark. For the single-view RGB-D input $I_{rgbd}$, we provide a dataset $\mathcal{V}_r$, which comprises RGB-D images of all the objects in $\mathcal{O}_s$. For the instruction input $\tilde{I}$, we construct a dataset of 70+ rotation-specified instructions, denoted as $\mathcal{I}_r$. For each object category in $\mathcal{O}_s$, we label it with 1-5 instructions in $\mathcal{I}_r$ based on its features. The instructions are categorized into 3 levels that progressively increase in difficulty. Level 0 includes basic instructions that are related to the geometric shape of the object, such as "upright" and "upside down". Level 1 generally requires a higher understanding of direction and orientation, such as "handle to the left". Level 2 contains

harder instructions concerning semantics and textual information of the object, such as "label forth" and "characters right side up".

**Evaluation metrics.** Due to the diversity and complexity of human commands in our instruction set $I_r$, it is difficult to design a uniform metric to judge alignment between the numerical representation of the rotation and the initial instruction. To address this problem, we manually annotate each task with a rotation range that complies with the instructions. Rotation results that fall into this range are considered as correct, otherwise wrong.

### 3.4. 6-DoF-track Benchmark

**Data composition and annotation.** The 6-DoF-Track Benchmark comprises 1059 tasks, providing a comprehensive evaluation that jointly assesses the rotation and position performance of an Open6DOR task. The formulation of the RGB-D scene image $I_{rgbd}$ aligns with that of the Position-track. For instruction $\tilde{I}$, we combine instructions from $\mathcal{I}_p$ and $\mathcal{I}_r$, forming instructions that specify both the position and rotation of an object. Each instruction is paired with an RGB-D scene image as the task input, and we exclude the incompatible pairs to ensure that the tasks are well-defined and performable.

**Evaluation metrics.** We evaluate the quality of a 6-DoF pose from two perspectives: rotation and position. Specifically, we manually annotate the desired rotation and position of the target object based on the instruction. We consider a task successful only when it satisfies both criteria.

## 4. Open6DOR-GPT

### 4.1. Method Overview

As shown in Fig. 2, we enhance GPT-4V[25]'s capabilities to address the challenges of the Open6DOR task in a decomposed way. Initially, the Task Preprocessing Module deciphers $\tilde{I}$ based on the $I_{rgbd}$ and feeds the resulting images to the Position Module and Rotation Module respectively. Within the two modules, we empower GPT-4V with 3D awareness and simulation assistance, thereby effectively outputting the predicted goal position $P_{\text{goal}}$ and rotation $R_{\text{goal}}$. Finally, the Simulation-assisted Planning Module identifies a suitable grasping pose and plans out an optimal action trajectory to accomplish the task. We will first introduce each module of our proposed system in paragraph *B-E* to explain how an Open6DOR task is accomplished. We then elaborate on how the system tackles long-horizon tasks with multiple rounds of operations.

### 4.2. Task Preprocessing Module

With the single-view RGB-D Image $I_{rgbd}$ and the task instruction $\tilde{I}$ as input, this module leverages GPT-4V to interpret the instruction and identifies object names $\{O_i^{name}\}$,

which in turn triggers GroundedSAM [18] to generate a set of labeled masks. Based on the masked Image $I_{mask}$, the RGB image of the target object $I_{object}$ is extracted. These images are used in subsequent modules.

### 4.3. 3D-aware Position Module

Taking the masked RGB-D image $I_{mask}$ and task instruction $\tilde{I}$ as input, the 3D-aware Position Module $\mathcal{M}_p$ determines and outputs the goal position that complies with the requirements.

To incorporate three-dimensional (3D) data into GPT-4V's understanding, our approach utilizes back-projection based on $I_{mask}$ to generate a 3D masked point cloud, symbolized as $PC_i^{3d}$. This computation includes determining the centroid $\text{Center}_i^{3d}$ and bounding box $\text{Bbox}_i^{3d}$ of the point cloud associated with the queried object.

$$PC_i^{3d} = \text{BackProj}(I_{rgbd}(\text{Mask}_i^{2d})) \tag{1}$$

$$\text{Center}_i^{3d}, \text{Bbox}_i^{3d} = \text{Mean}(PC_i^{3d}), \text{Max}(PC_i^{3d}) - \text{Min}(PC_i^{3d})$$

These spatial attributes are then integrated back into the prompt for GPT-4V, facilitating the model to accurately ascertain the goal position for the target object $P_{\text{goal}}$.

### 4.4. Simulation-assisted Rotation Module

With the single-view RGB image of the target object $I_{object}$ and the task instruction $\tilde{I}$ as input, the rotation module would output the goal rotation $R_{\text{goal}}$ for the object. We first reconstruct the target object from $I_{object}$ using One-2-3-45++ [22], and outputs a textured mesh, denoted as $M$. The reconstruction process is followed by four phases: (1) large-scale sampling (2) simulation-assisted filtering (3) rotation categorization (4) GPT-4V selection.

**Large-scale sampling.** In Phase 1, we randomly sample a total amount of $N$ rotations $\{R_i^0\}_{i=0}^N$ as initial inputs for subsequent phases. We set $N = 3600$ and use Uniform Sampling in SO(3) space (Special Orthogonal Group in 3D space) to ensure the diversity of our samples.

**Simulation-assisted filtering.** Now that we have a large pool of rotation candidates $\{R_i^0\}$, the goal of Phase 2 is to filter out the unreasonable candidates and narrow down the sample pool efficiently. To accomplish this task, we first examine the stability of the rotation candidates by incorporating a physics simulator through which all the unstable poses are excluded. To be specific, each $R_i^0$ is applied to a replica of $M$, denoted by $M_i$, as its initial rotation in the simulator, amounting to $N$ actors $\{M_i^0\}_{i=0}^N$ in total. Then, all the actors are dropped from a low height, landing on the ground with diverse ending poses. We record the relative rotation from the original mesh $M$ to the ending pose of $M_i$ as $R_i^t$. By now, we have narrowed down the originally random

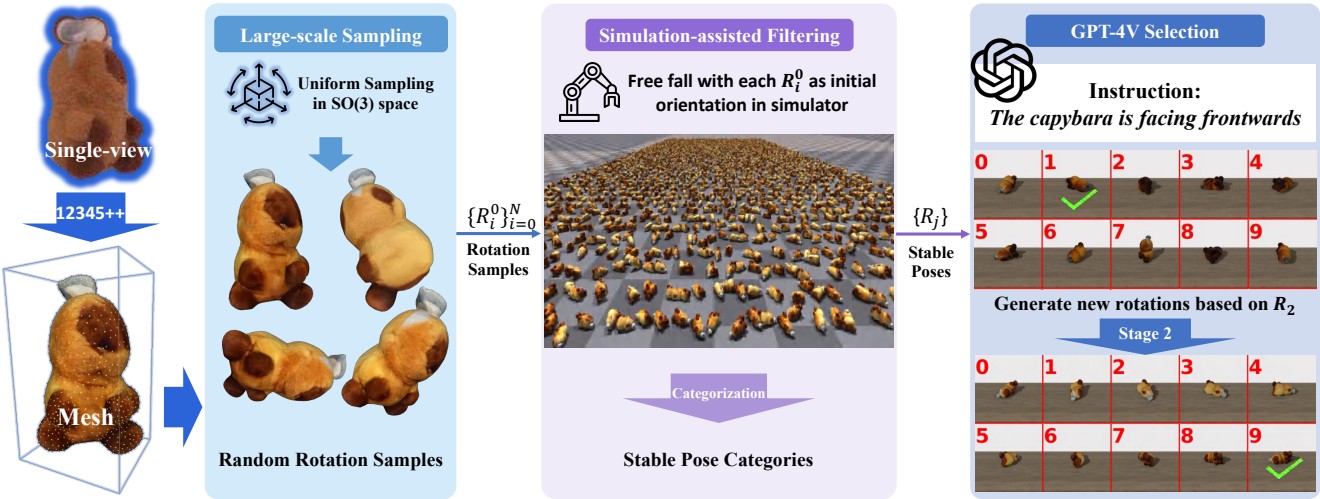

Figure 3. **Simulation-assisted Rotation Module.** Firstly, a textured mesh is reconstructed from the single-view image of the target object. Then, we employ large-scale sampling to obtain multiple rotation samples. This sample set is then narrowed down through a simulation-assisted filtering process to derive several stable pose categories. Finally, we generate rendered images of the pose candidates, from which GPT-4V selects the optimal goal rotation.

and irregular distribution of $\{R_i^0\}_{i=0}^N$ to a more condensed space $\{R_i^t\}_{i=0}^N$ of geometrically stable rotations. By simulating the dropping process within 10 seconds, we avoid the time-consuming inferences of vision language models while accurately extracting stable poses of $M$.

**Rotation categorization.** From Phase 2, we've obtained a set of rotations $\{R_i^t\}_{i=0}^N$ that guarantee the stability of the object. However, these rotations are unevenly cluttered around several distinct centers, each representing a stable pose. In order to categorize these rotations and extract a representative for each stable pose, we propose a criterion by which we judge whether two rotations $R_a$ and $R_b$ could be classified into the same category. In brief, we regard $R_a$ and $R_b$ as identical if they meet any one of the following two criteria: (1) the relative rotation from $R_a$ to $R_b$ is small in magnitude (2) $R_a$ and $R_b$ represents symmetrical poses that are transferrable via a rotation along the z-axis(perpendicular to the table surface). We define a threshold for each of these criteria and calculate whether $R_a$ and $R_b$ could be considered as identical. Through this method, we classify the rotations into several clusters and represent each category with a single rotation, which largely reduces the total number of rotation candidates for the task. Therefore, we successfully narrow down $\{R_i^t\}_{i=0}^N$ to a small set: $\{R_j\}_{i=0}^n$, in which each rotation represents a distinct stable pose.

**GPT-4V selection.** During the last phase of our rotation engine, we feed the filtered set of rotations $\{R_j\}_{i=0}^n$ in the form of the 2D image along with the original instruction to GPT-4V and let it select a candidate as the goal rotation $R_{\text{goal}}$. To transform $\{R_j\}$ into a modal that VLM could easily understand, we apply each $R_j$ to $M$ and render the image of the object loaded on a table accordingly. The im-

ages are then arranged together into a collage, with an index mark on the upper left of each grid. Empirically, we found that this strategic approach of numbering and segmenting the images boosts the performance of GPT-4V in selecting the right answer. To further enhance our method, we employ a two-stage strategy that resamples a set of rotation candidates based on the rotation GPT-4V has selected in Stage 1. After this second round of adjustment, the goal rotation $R_{\text{goal}}$ is determined and outputted.

## 4.5. Simulation-assisted Planning Module

Utilizing the predicted goal position $P_{\text{goal}}$ and goal rotation $R_{\text{goal}}$, the planning module formulates an effective execution strategy with simulation assistance. Firstly, the Grasp Detection Model, GSNet [30], takes the refined point cloud $PC_{\text{refined}}$ as input and generates a series of scored grasping pose candidates $\{(G_j, s_j)\}$. From $\{G_j\}$, GPT-4V selects valid grasping poses that rest on the target object by leveraging the object bounding box $\text{Bbox}^{3d}$ derived from the 3D-aware Position, resulting in a ranked set of $\{\tilde{G}_j\}$.

$$\{(G_j, s_j)\} = \text{GSNet}(PC_{\text{refined}}) \tag{2}$$

$$\{\tilde{G}_j\} = \text{Sorted}_s(\{(G_j)\}_{\text{Bbox}^{3d}}) \tag{3}$$

Next, we use cuRobo[29] as the motion planner, which enumerates $\{\tilde{G}_j\}$ within the simulator based on their score rankings $\{s_j\}$, and identifies a trajectory that optimizes both grasping and placement, denoted as $\mathcal{T}$. Finally, the robot employs its control system to accomplish execution according to $\mathcal{T}$.

| Success Rate (%) | Level 0 | Level 1 | Level 2 | Overall |
|---|---|---|---|---|
| GPT-4V[25] | 45.1 | 40.3 | 50.0 | 44.5 |
| Dream2Real* [20] | 19.9 | 32.6 | - | 23.5 |
| VoxPoser*[17] | 23.6 | 19.4 | 0.0 | 19.6 |
| VoxPoser(VLM)*[17] | 27.1 | 27.9 | 0.0 | 23.9 |
| **Open6DOR-GPT** | **83.7** | **62.7** | **80.0** | **78.0** |

Table 2. **Results on Position-track Benchmark.** We compared our approach against several benchmarks for positioning proposals. This includes: (1) GPT-4V[25], utilizing pixel input to predict object placement and employing depth for 3D location. (2) A tailored Dream2Real[20] baseline for our task. (3,4) VoxPoser[17] original and adapted versions, aligning with our goals. Our tests include GPT-4V's Large Language Model (LLM) and Vision-Language Model (VLM) setups, with an asterisk denoting ground-truth data usage as reference baselines.

## 4.6. Long-horizon Open6DOR Execution

With the framework outlined in previous sections, our system is capable of managing individual Open6DOR tasks. For long-horizon rearrangement tasks, we first employ GPT-4V to evaluate whether multiple Open6DOR steps are required. If the task necessitates multiple steps, GPT-4V leverages high-level planning to divide the instruction $\tilde{I}$ into several discrete execution steps $\{\tilde{I}_i\}$. Subsequently, for each individual step $I_i$, we apply the previously described methodology to carry out the task. Upon the completion of each Open6DOR sequence, the overall task is considered complete. This approach ensures a systematic and efficient handling of complex rearrangement tasks, breaking them down into manageable steps that are executed with precision.

## 5. Experiments

### 5.1. Results on Position-track Benchmark

We evaluate the performance of our position module and several baselines on the Position-track Benchmark. As shown in Table 2, both Dream2Real and GPT-4V demonstrate incompetence at precise position determination. VoxPoser[17], another baseline, yields unsatisfactory performance due to reliance on Large Language Models (LLM) without visual inputs. But even when adapted to a VLM-assisted version and incorporating image data, VoxPoser(VLM)* fails to gain significant improvements. Comparatively, our approach markedly surpasses all these baselines by over 30 percent, demonstrating superior and consistent performance on the Position-track Benchmark.

### 5.2. Results on Rotation-track Benchmark

Our Rotation Module comprises four phases aimed at enhancing GPT-4V[25] through a simulation-assisted sample-

| Success Rate(%) | Level 0 | Level 1 | Level 2 | Overall |
|---|---|---|---|---|
| GPT-4V[25] | 9.1 | 6.9 | 11.7 | 9.2 |
| Dream2Real*[20] | 37.3 | 27.6 | 26.2 | 31.1 |
| S-F + GPT-4V | 49.0 | 32.7 | 28.1 | 38.0 |
| **Open6DOR-GPT (S-F + 2-Stage 4V)** | **50.3** | **36.4** | **41.8** | **43.7** |

Table 3. **Results on Rotation-track Benchmark.** Quantitative comparison with a refined version of Dream2Real[20] method (replacing CLIP Model with GPT-4V), and ablation studies of different phases in the Rotation Module. "S-F" stands for "Sampling-Filtering". The first three rows ablate Phase1-4, Phase3-4, and Stage2 in Phase 4, respectively.

| Success Rate (%) | Rotation | Position | Overall | Time Cost |
|---|---|---|---|---|
| Dream2Real[20] | - | - | - | >700s |
| Dream2Real*[20] | 18.7 | 26.2 | 13.5 | 358.3s |
| **Open6DOR-GPT** | **52.1** | **78.0** | **40.6** | 156.3 s |

Table 4. **Results on 6-DoF-track Benchmark.** We compare our method with an optimized version of Dream2Real[20] on the 6DoF Benchmark (denoted as Dream2Real*). The three columns depict the quality of the goal pose in terms of rotation, position, and overall performance.

and-filter mechanism. To evaluate the effectiveness of each phase, we conduct ablation studies using the Rotation-track of Open6DOR Benchmark, with results detailed in Table 3. We compare our approach with Dream2Real[20], replacing their CLIP Model with GPT-4V to ensure fairness. As shown in the first row of the table, directly querying GPT-4V yields an unsatisfactory success rate. Substituting our module with Dream2Real's method also leads to a noticeable performance decline. However, upon incorporating the Simulation-assisted Filtering Phase, we observe a noticeable performance increase as GPT-4V is able to choose from a confined set of rotation candidates. Further enhancements are achieved by integrating a 2-stage VLM Selection method. Notably, the combined module outperforms Dream2Real* by 12 percent.

### 5.3. Results on 6-DoF Benchmark

We evaluate our entire pipeline using the 6DoF-track of Open6DOR Benchmark. The evaluation of rotational, positional, and joint performance are presented in Table 4. For baseline methods, we found a limited number of works addressing the 6DoF problem and chose to compare with Dream2Real[20]. However, the original Dream2Real method is excessively time-consuming, requiring over 10 minutes per task for completion. To fully assess Dream2Real on the 6DoF-track Benchmark (con-

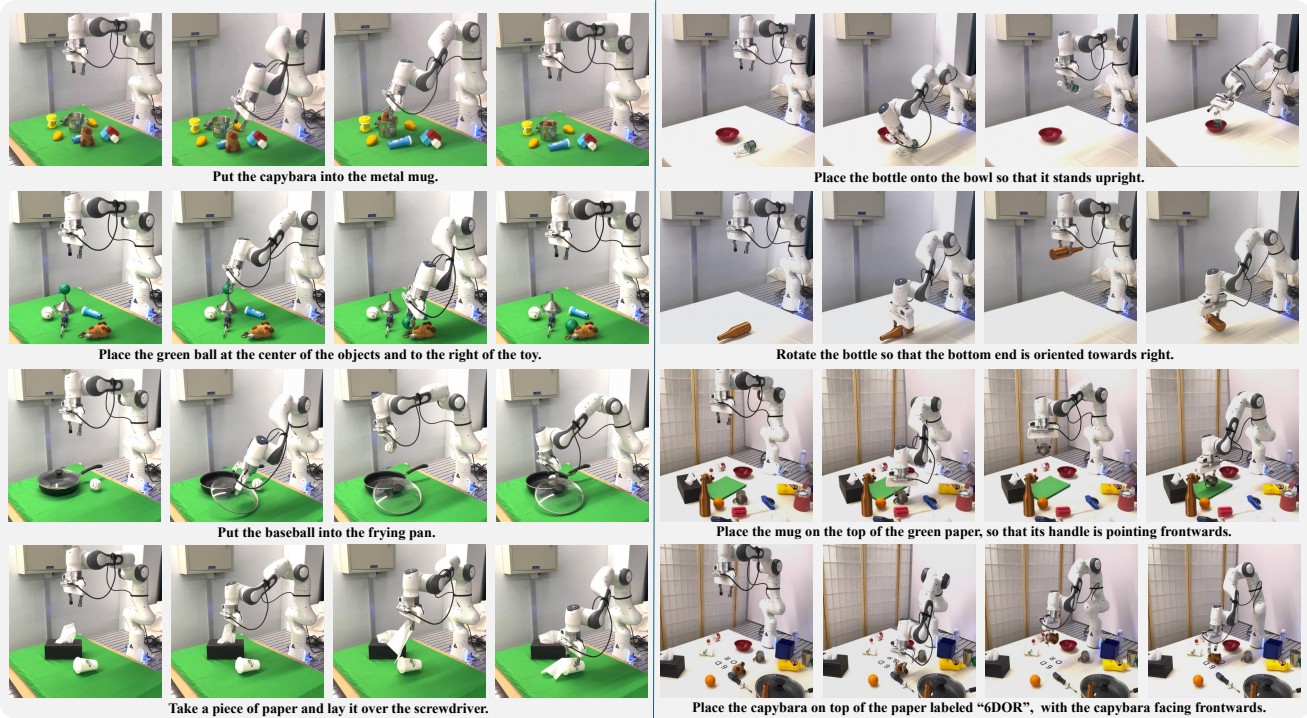

Figure 4. **Real-world Experiments.** We ground Open6DOR-GPT in real-world settings and conduct various tasks as well as long-horizon highlighting its exceptional zero-shot generalization capability across challenging tasks.

taining 1k+ tasks), we skip the scene-scanning and reconstruction section of their method, using ground-truth mesh and image instead. Despite the disadvantage of the mesh quality on our side(our method reconstructs a mesh from the original image), Open6DOR-GPT significantly outperforms Dream2Real* by about 30 percent. Our approach also demonstrates better efficiency compared to baseline approaches.

## 5.4. Real-world Experiments

In our real-world experiments, we leverage a Franka Panda arm with a parallel gripper and mount a Realsense D415 camera to its end for image capturing. To comprehensively demonstrate the performance of our approach, we design tasks of varying difficulty levels: (1) place objects to the target position (2) place objects to the target rotation (3) place objects to the target position and rotation. We employ diverse objects with different geometries, textures, and materials, including transparent and specular ones. As shown in Fig. 4, our zero-shot method is able to tackle challenging Open6DOR scenarios and demonstrates strong potential in long-horizon tasks.

## 6. Conclusion

In this paper, we pioneer the establishment of the Open6DOR benchmark and VLM-based approach, addressing the need for a comprehensive evaluation and a foregoing

method exploration in open-instruction 6-DoF object rearrangement. Our synthetic benchmark, comprising over 200 objects and 2400 tasks, offers a standardized framework for evaluating the capabilities of embodied agents in simulation environments. Additionally, our Open6DOR-GPT approach achieves state-of-the-art performance, augmenting GPT-4V with 3D awareness and simulation assistance. As for the current limitations, while Open6DOR-GPT significantly improves position and rotation handling, it does not achieve real-time performance, and rotation understanding remains suboptimal. Future enhancements to our benchmark are anticipated, especially real-world extensions.

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
