# OpenReview forum: "Open6DOR: Benchmarking Open-instruction 6-DoF Object Rearrangement and A VLM-based Approach"
_thecvf.com/CVPR/2024/Workshop/VLADR — VLADR 2024 Oral_

### Official Review · Reviewer_dqZe · 2024-04-17

**Rating:** 7
**Confidence:** 4

**Review:**

This paper systematically studies and benchmarks the performance of existing VLMs for Open-instruction 6-DoF object rearrangement. I tend to accept the paper because:

- This benchmark is timely as the community is paying more attention to language-guided robot manipulation tasks.
- The proposed Open6DOR-GPT surpasses other baselines significantly.
- It is highly relevant to our workshop.

However, authors should revise the paper carefully to avoid obvious flaws such as the template-named title.

---

### Decision · Program_Chairs · 2024-04-22

Accept (Oral)